# Selenol-Based Nucleophilic Reaction for the Preparation of Reactive Oxygen Species-Responsive Amphiphilic Diblock Copolymers

**DOI:** 10.3390/polym11050827

**Published:** 2019-05-08

**Authors:** Xiaowei An, Weihong Lu, Jian Zhu, Xiangqiang Pan, Xiulin Zhu

**Affiliations:** Jiangsu Key Laboratory of Advanced Functional Polymer Design and Application, College of Chemistry, Chemical Engineering and Materials Science, Soochow University, Suzhou 215123, China; 20164009027@stu.suda.edu.cn (X.A.); luweihong@suda.edu.cn (W.L.); xlzhu@suda.edu.cn (X.Z.)

**Keywords:** RAFT, selenol, amphiphilic polymer, drug delivery

## Abstract

Selenide-containing amphiphilic copolymers have shown significant potential for application in drug release systems. Herein, we present a methodology for the design of a reactive oxygen species-responsive amphiphilic diblock selenide-labeled copolymer. This copolymer with controlled molecular weight and narrow molecular weight distribution was prepared by sequential organoselenium-mediated reversible addition fragmentation chain transfer (Se-RAFT) polymerization and selenol-based nucleophilic reaction. Nuclear magnetic resonance (NMR) and matrix-assisted laser desorption/ionization time-to-flight (MALDI-TOF) techniques were used to characterize its structure. Its corresponding nanomicelles successfully formed through self-assembly from the copolymer itself. Such nanomicelles could rapidly disassemble under oxidative conditions due to the fragmentation of the Se–C bond. Therefore, this type of nanomicelle based on selenide-labeled amphiphilic copolymers potentially provides a new platform for drug delivery.

## 1. Introduction

Compared with sulfur, selenium shows versatile properties owing to its larger atomic radius and relatively lower electronegativity [1]. Selenium-containing polymers have attracted a great deal of attention in recent years and have been widely used as photoelectric materials, adaptive materials, and biomedical materials [2,3]. Selenophene polymers have been considered to be effective photoelectric materials which may be widely used in solar cells, molecular switches, thin film transistors, etc. [4,5,6,7,8,9,10,11,12]. Diselenide-containing adaptive materials were successfully incorporated in the fabrication process under very mild conditions to achieve self-healing properties [13,14,15,16]. Selenium-containing polymers show versatile responsive behaviors to multiple stimuli, such as oxidation, reduction, and irradiation [17,18,19,20,21,22,23], which makes them potentially useful bio-building blocks. Traditional methods for preparing selenium-containing polymers used step growth polymerization [24,25,26,27,28], radical polymerization [29,30,31,32,33,34], and ring-opening polymerization [35,36,37], and our group successfully developed organoselenium-mediated controlled radical polymerization (Se-RAFT) to prepare selenium-containing polymers with both controlled molecular weight and narrow molecular weight distribution [38]. Subsequently, the application of selenol-based nucleophilic substitution and Se-Michael addition reactions for polymer chain end modification was presented [39], so that many functional groups could be introduced to the selenium-containing polymers. Herein, we reported a new application of selenol-based nucleophilic reaction for the design of a reactive oxygen species-responsive amphiphilic diblock copolymer. Firstly, the diselenocarbonate-end capped polystyrenes with different molecular weights (MWs) were prepared through the organoselenium-mediated radical polymerization of styrene (St). After aminolysis of diselenocarbonate with hexylamine, nucleophilic attack of the exposed selenol to poly(ethylene glycol) methyl ether methacrylate (PEGMA) gave a selenide-labeled diblock copolymer (denoted as PS-Se-b-PEGMA; see Scheme 1). PS-Se-b-PEGMA is able to form micellar aggregates in water which are disassembled in oxidation conditions.

## 2. Experimental Section

### 2.1. Materials

Styrene was purchased from Shanghai Chemical Reagents Co. Ltd., Shanghai, China, and purified before use. PEGMA (*M*_n_ = 300, 500, and 1000 g mol^−1^; Aldrich) was passed through an alumina column to remove the inhibitor. After that, it was dried with calcium hydride, then distilled under reduced pressure and kept in a refrigerator below 0 °C. 2,2’-Azoisobutyronitrile (AIBN, 98%) was recrystallized from ethanol and then stored in a refrigerator at 4 °C. Se-benzyl O-(4-methoxyphenyl) carbonodiselenoate (Se-CTA) was synthesized according to a previously reported method [33]. Tributylphosphane (Bu_3_P, 98%) was purchased from Adamas Reagent Ltd, Shanghai, China. Dialysis bags (molecular weight cutoff: 1000 Da) were purchased from Sinopharm Chemical Reagent Co. Ltd., Shanghai, China. Tetrahydrofuran (THF), N,N-dimethylformamide (DMF), methanol (MeOH), and other chemicals were purchased from Shanghai Chemical Reagents Co. Ltd. Shanghai, China and used without further treatment. Doxorubicin hydrochloride (DOX·HCl, 99%) was purchased from Sigma, Shanghai, China.

### 2.2. Characterization

The number-average molecular weight (*M*_n_) and molecular weight distribution (*Ð*) of the resulting polymers were determined by a TOSOH HLC-8320 size exclusion chromatograph (SEC) equipped with a TSKgel SuperMultiporeHZ-N column (3) (4.6 × 150 mm) at 40 °C. Tetrahydrofuran served as the eluent with a flow rate of 0.35 mL min^−1^. SEC samples were injected using a TOSOH HLC-8320 SEC plus autosampler. The molecular weights were calibrated with polystyrene (PS) standards. ^1^H (300 MHz) NMR spectra were recorded on a Bruker Avance 300 spectrometer. Chemical shifts are presented in parts per million (*δ*) relative to CHCl_3_ (7.26 ppm in ^1^H NMR). Transmission electron microscopy (TEM) was performed with a HITACHI HT7700 microscope operating at a 120-kV accelerating voltage. The fluorescence emission spectra (FL) were obtained on a HITACHI F-4600 fluorescence spectrophotometer at room temperature. Hydrodynamic diameter (*D*_h_) was determined by dynamic light scattering (DLS) using a Brookhaven NanoBrook 90Plus PALS instrument at 25 °C with a scattering angle of 90°. Fourier transform infrared spectroscopy (FT-IR) was recorded with the Bruker TENSOR 27 FT-IR instrument using the conventional KBr pellet method. The elemental composition of the surfaces was measured with X-ray photoelectron spectroscopy (XPS) (Thermo Fisher Scientific ESCALAB 250 XI, Al KR source).

### 2.3. Typical Procedure of Organoselenium-Mediated Controlled Radical Polymerization

A dry 10-mL ampule was filled with styrene (8.0 mL, 80 mmol), Se-benzyl *O*-(4-methoxyphenyl) carbonodiselenoate (Se-CTA) (0.6147 g, 1.6 mmol), and AIBN (0.0788 g, 0.48 mmol). The solution was degassed using the standard freeze–pump–thaw method (at least 3 cycles). The ampule was flame-sealed and placed into an oil bath, which was thermoset at the desired temperature. At timed intervals, the ampule was immersed into iced water and then opened. The contents were dissolved in 3 mL of tetrahydrofuran (THF) and precipitated into 400 mL of methanol. Carbonodiselenoate-labeled polystyrene (PS-Se) was obtained by filtration and then dried to constant weight at room temperature under vacuum. The conversion of styrene was gravimetrically determined.

### 2.4. Typical Procedure of Selenol-Based Nucleophilic Addition of PS-Se to PEGMA

Without additional notes, a typical procedure for optimization of reaction time is given below as an example. A dry 5-mL ampule was filled with PS-Se-1 (*M*_n,SEC_ = 1900 g mol^−1^, 95 mg, 0.059 mmol), PEGMA950 (*M*_n_ = 950 g mol^−1^, 0.59 g, 0.59 mmol), and DMF (2.0 mL) with a stir bar. After being thoroughly bubbled with argon for 15 min to eliminate the dissolved oxygen, n-hexylamine (30 μL) was added. Then, the ampule was flame-sealed immediately. After stirring for 1 d at 60 °C, the solution was precipitated into 100 mL of methanol. The polymer was obtained by filtration and then dried to constant weight at room temperature under vacuum.

### 2.5. Fabrication of PS-Se-b-PEGMA Micelles

PS-Se-b-PEGMA (2 mg) was dissolved in DMF (2 mL), and then deionized water (0.3 mL) was added to the solution using a syringe pump at the rate of 0.2 mL h^−1^ at room temperature. After addition of the water, the suspension was stocked for 1 day to stabilize the aggregates. Then, the suspension was dialyzed in a dialysis bag (molecular weight cutoff: 1000 Da) against deionized water for at least 24 h to remove DMF. After dialysis, the suspension was added to deionized water until the volume increased to 2 mL with a concentration of 1 mg mL^−1^ for further tests.

### 2.6. Chemical Oxidation of PS-Se-b-PEGMA Micelles by H_2_O_2_

In brief, 0.5 mL of PS-Se-*b*-PEGMA micelle suspension was kept in a 1-mL ampule and then placed into H_2_O_2_ solution (33 mM). After stirring for 5 h, the micelle suspension was freeze-dried for TEM studies.

### 2.7. In Vitro Cytotoxicity Study of the Micelles

The samples were disinfected under ultraviolet light, and then five groups of extracts (2, 4, 6, 8, and 10 mg mL^−1^) were prepared. The blank group (culture medium) and the control group (culture medium and cells) were set up to compare with the experimental group. NIH-3T3 cells were inoculated on 96-well plates at a density of 8 × 10^4^ mL^−1^ (8 × 10^3^ well^−1^). Cells were cultured in incubators (at 37 °C and 5% carbon dioxide) to become adhered to the 96-well plates. After 24 hours, the medium was removed and the extract was added to culture. Then, 24 h later, the extract was removed and 10 μL CCK-8 solution and 90 μL culture medium were added to each pore. Then, cells were incubated at 37 °C for 1 h. A microplate reader was used to measure the absorbance of the sample at 450 nm.

### 2.8. Drug Loading

The following process was carried out in the dark. A mixture of 5.0 mg DOX·HCl and 3.6 μL of triethylamine (TEA) in 1.0 mL of dimethyl sulfoxide (DMSO) was added to a 2-mL ampule. After stirring overnight, excess TEA was removed by rotary evaporation to give the hydrophobic DOX solution. Then, 4 mL of PBS solution (50 mM, pH 7.4) was added dropwise to the mixture of copolymer in DMF (1.0 mL, 1 mg mL^−1^) and DOX base solution (50 μL, 5.0 mg mL^−1^) with stirring at room temperature. Afterwards, in order to remove both unencapsulated DOX and the organic solvent, the mixture was dialyzed against PBS solution (50 mM, pH 7.4) for 24 h.

The amount of DOX was determined by fluorescence (FL4600) measurement (excitation at 480 nm). First, a calibration curve was obtained by measuring the fluorescence intensity of different concentration DOX/DMSO solutions. Second, the fluorescence intensity of DOX-loaded micelles dissolved in DMSO was analyzed. The amount of DOX loaded in the micelles could be determined using the calibration curve.

The drug loading content (DLC) and drug loading efficiency (DLE) were calculated using the following formulas:

DLC (wt.%) = (weight of loaded drug / weight of (polymer + loaded drug)) × 100%

DLE (wt.%) = (weight of loaded drug / weight of drug in feed) × 100%

### 2.9. Oxidation-Responsive Drug Release

In brief, 33 mM H_2_O_2_ was added into freshly prepared self-assembled solution (1.0 mL) of PS-Se-1-*b*-PEGMA_950_ in PBS. The reaction mixture was stirred at 25 °C for 3 h. 

Fluorescence spectrophotometry was then used to monitor the change in fluorescence intensity of the micelle solution.

## 3. Results and Discussion

### 3.1. Organoselenium-Mediated Controlled Radical Polymerization

The diselenocarbonate-end capped polymers were prepared through the organoselenium-mediated polymerization of styrene (St) according to our previous reports [33]. Two polymers, PS-Se-1 and PS-Se-2, with different molecular weights and narrow molecular weight distribution (<1.20) were prepared, as detailed in Table 1. The structure of PS-Se-1 was characterized by ^1^H NMR. As shown in Figure 1, the signals at *δ* 3.78 ppm (3H, *I*_3.78_ = 3.00) were ascribed to the protons of the methoxy group (–OMe), and the signals around *δ* 4.50 ppm (1H, *I*_4.43-4.68_ = 1.06) were ascribed to the protons of CH-Se. The signals at around δ 7.00 ppm (5H, *I*_6.60-7.11_ = 92.1) were ascribed to the protons of the phenyl group. Thus, the molecular weight (*M*_n, NMR_) can be calculated to be 2100 g mol^−1^ by the equation *M*_n, NMR_ = 104 × [(92.1 − 9) / 5] + 91 + 295. The polymers were also measured by using SEC with coupled refractive index (RI) and UV detectors (Figure 2). The two curves almost coincide, and the molecular weight (*M*_n,GPC_ = 1900 g mol^−1^) was close to the value obtained by ^1^H NMR. All evidence proved the high chain end functionality of the diselenocarbonate-ended polystyrene (PS), which ensured further chain end modification. 

### 3.2. Reaction Condition Optimization of Selenol-Based Nucleophilic Reaction

As in our previous reports, diselenocarbonate was expected to be converted to selenol by amine compounds [39]. In the present work, after the rapid aminolysis of PS-Se-1 (*M*_n_, _SEC_ = 1900 g mol^−1^) by *n*-hexyl amine, the corresponding selenol reacted with PEGMA_950_ (*M*_n_ = 950 g mol^−1^) to make the block copolymers (Scheme 2). We initiated our studies by examining the effect of time on selenol-based nucleophilic reaction. The results are presented in Table 2. Screening experiments indicated that moderate conversion of PS-Se-1 could be obtained after four days (entries 1, 2, 3, and 4). An increase in temperature from 25 °C to 60 °C resulted in the increase of the conversion of PS-Se-1 from 22.4% to 50% (Table 2, entry 5). It was already found that Bu_3_P could act as a reducing agent to prevent oxidative coupling of selenol. Without Bu_3_P, the conversion of PS-Se-1 dropped to 14.6% (Table 2, entry 6). Bu_3_P also acts as a catalyst for the subsequent selenol-Michael addition reaction. Consequently, no more catalyst, such as DBU and Et_3_N, was needed to be added to this system (entries 7 and 8) [1]. Lastly, the effect of molar ratio on selenol-based nucleophilic reaction was examined. The reaction became more smooth as the amount of PEGMA (entries 9 and 10) increased. When the molar ratio of PS-Se-1:PEGMA = 1:10, the reaction reached the highest conversion rate of about 85% after separation. In contrast, the higher molar ratio of PS-Se-1:PEGMA (1:50) resulted in an extremely viscose solution, which may prevent further reaction. Moreover, purification loss also decreases the product yield.

### 3.3. Selenol-Based Nucleophilic Reaction of PS-Se-1 and PEGMAs with Different Molecular Weights

After studying reaction condition optimization for selenol-based nucleophilic reaction of PS-Se-1 with PEGMA_950_, PEGMAs with different molecular weights were examined. The results are listed in Table 3. PS-Se-1 reacted efficiently with PEGMA and the conversion rate of PS-Se-1 with PEGMA_320_ peaked at 95.4%, and the conversion rate of PS-Se-1 decreased with the increase of molecular weight of PEGMA. The *M*_n_ of copolymers was determined by SEC. As shown in Figure 3, the curves shifted significantly after the nucleophilic reaction, which evidenced the successful modification of PEGMA at the end of PS-Se-1. In the ^1^H NMR spectra, the proton signals of the corresponding vinyl shifted from 6.10 and 5.54 ppm to 2.57 and 2.49 ppm, and the proton signals of CH groups close to Se shifted from 4.70 and 4.47 ppm to 2.84 ppm, indicating a complete conversion of PEGMA (Figure 4). Also, MALDI-TOF mass spectrometry was used to further characterize the structure of the copolymer. As shown in Figure 5, the main population at the isotropic peak at 2340.428 m/z matched the theoretical calculation well (2340.152 m/z). Furthermore, two main sequence peaks are very close to the styrene (104.15 g mol^−1^) and CH_2_CH_2_O (44.03 g mol^−1^) units. All the evidence indicated the efficiency of this reaction.

### 3.4. Self-Assembly of PS-Se-1-b-PEGMA_950_ Before and After Oxidation

Selenium-containing copolymers have shown redox responsiveness in many systems. Some selenide-containing aggregates can respond rapidly to external redox stimuli and subsequently release the incorporated species under mild conditions [26,28]. Here, self-assembly behavior of the three copolymers were investigated. It was found that PS-Se-1-*b*-PEGMA_950_ could self-assemble spontaneously in aqueous solution through hydrophobic/hydrophilic interaction. As shown in Figure 6, the TEM measurement of PS-Se-1-*b*-PEGMA_950_ micelles showed spherical aggregates with an average diameter of 30 nm. The DLS results are in accordance with the TEM results with an average diameter of 56 nm (Figure 7a). It is noteworthy that the micelles were quite stable in an ambient environment and could maintain their structures for at least one month. Then, H_2_O_2_ solution was used as the oxidant to study the oxidation responsiveness of the selenide block copolymer aggregates. From the TEM images shown in Figure 6c,d, the micellar structure was converted to irregular aggregates after two hours of oxidation process, and these irregular aggregates were further decomposed for another three hours. The DLS results also proved that the morphology change of the aggregates occurred after adding H_2_O_2_ solution. Furthermore, XPS was also used to analyze these micelles before and after oxidation treatment. As shown in Figure 7b, the binding energy of Se 3d^5^ shifts from 56.22 eV to 60.81 eV, suggesting a higher valency of selenium which is close to the seleninic acid group [21]. All the results proved that oxidative cleavage of the selenide linkage leads to the morphology change of the micelle. 

### 3.5. Cytotoxicity Test

The nanomicelles based on selenide-labeled amphiphilic copolymers potentially provide a new platform for targeted drug delivery. We examined the cytotoxicity of PS-Se-1-*b*-PEGMA (*M*_n,SEC_ = 3700 g mol^−^^1^, *Ð* = 1.11, 1 × 10^−^^4^ M). As shown in Figure 8, it can be seen that the PS-Se-*b*-PEGMA showed low cytotoxicity when compared with other selenide-containing polymers [40]. The obvious inhibitory effect on the NIH-3T3 cells was shown when the concentration of the micelles was high, at up to 1.6 mg mL^−1^.

### 3.6. Drug Loading and Oxidation-Responsive Drug Release

The anticancer drug doxorubicin (DOX) was chosen as a model molecule for encapsulation. The self-assembly of PS-Se-1-*b*-PEGMA_950_ (1 mg mL^−1^) was conducted in DMF solution in the presence of DOX (0.1 mg mL^−1^), which has a characteristic maximum emission at 590 nm. The DOX-loading micelles were purified with dialysis membrane, and the DOX concentration was calculated by the fluorescence emission spectra. The drug loading content (DLC) was evaluated to be about 1.5%, and drug loading efficiency (DLE) was evaluated to be about 13.3%. Oxidation-triggered drug release studies in vitro were investigated at pH 7.4 and 25 °C by using H_2_O_2_. After oxidation for a specific time, the fluorescence emission of DOX was monitored, as shown in Figure 9a. The fluorescence intensity of DOX at 590 nm increased gradually, owing to the release of hydrophobic DOX from the drug carriers. The percentage release was calculated based on the fluorescence changes at 590 nm. The percentage release profile was plotted as a function of time, as shown in Figure 9b. 

## 4. Conclusions

In conclusion, a straightforward protocol for the synthesis of an oxidation-sensitive selenide-containing block copolymer has been developed on the basis of Se-RAFT. The diselenocarbonate termini of the polymers were readily converted to the selenide-containing block copolymer via aminolysis and selenol-based nucleophilic reaction. The present controlled radical polymerization (CRP)-based protocol offers a facile and straightforward fabrication of selenide-containing block copolymers that feature many monomer accessibilities, predictable MWs, and programmable polymeric architectures. These copolymers formed micelle-like nanoparticles of ~56 nm in diameter that show insignificant cytotoxicity of the micelles at high concentration and could be disrupted by H_2_O_2_. The biocompatibility and targeted cytotoxicity results suggest that these oxidation-sensitive selenide-containing block copolymers could be developed for controllable drug release.

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
