# Peer review of "Selenol-Based Nucleophilic Reaction for the Preparation of Reactive Oxygen Species-Responsive Amphiphilic Diblock Copolymers"

_polymers, 2019, doi:10.3390/polym11050827_

Reviewer 1 Report

In this study Pan, Zhu and coworkers have prepared Selenide-containing amphiphilic copolymers have shown significant potential for application in drug release system. The approach is interesting, however in its current form this work is not ready for publication.

As the authors describe in the introduction, step growth and  ring-opening polymerization

 cannot be precisely controlled but also the proposed approach cannot be defined as ”precise” synthesis of block copolymers. I agree with the low value of polydispersity for the (low Mn) first block containing Se  also if the conversion is not excellent. In my opinion the problems start with the growth of the second block where the conversion drops down to 85% with PEGMA 950 (the more interesting sample). I think that with this approach is really hard to get a clean system100% di-block copolymer that, I suppose, is required for applications in drug release.

-What happens with PEGMA 2000? Unfortunately, the authors explored only low Mn systems.

-Concerning the oxidation responsiveness: Please aurgue how the chosen oxidant (H2O2) is somehow related with the oxidant power expected during processes of drug release?

- The oxidation process should be proved also with other techniques (maybe NMR) not only via morphology

Author Response

Dear Reviewer,

         Thank you very much for giving us a chance to improve the quality of our manuscript entitled “Selenol-based Nucleophilic Reaction for Reactive Oxygen Species-Responsive Amphiphilic Diblock Copolymer”. We also thank you for your critical and professional comments for improving the quality of the manuscript. We have made specific efforts to perform additional experiments according to the comments/suggestions, and the manuscript has been revised adequately based on the supplemental results and a careful literature survey. The questions put forward by the reviewers have been answered point to point below.

Sincerely

Xiangqiang Pan

Reviewer 2 Report

In this work, the authors developed a methodology for the design of reactive oxygen species-responsive selenide-labeled amphiphilic di-block copolymer. This developed copolymer can self-assembled into micelles. This work is interesting, however, revision is still recommended. Please see the comments below.

(1) Why did the author only characterize PS-Se-1 and use PS-Se-1 for making the micelle? Why not use PS-Se-2?

(2) What are the size of the micelles? Please provide size range data using dynamic light scattering characterization.

(3) The TEM image data are not sufficient for indicating the oxidation stimuli responsiveness of these micelles. Please show quantitative data such as size changes or turbidity changes upon oxidation stimuli.

(4) The authors mentioned the micelles can be used for controlled drug release. Please add about the in vitro cytotoxicity study of the micelles and controlled drug release kinetics in vitro.

Author Response

Dear Reviewer,

         Thank you very much for giving us a chance to improve the quality of our manuscript entitled “Selenol-based Nucleophilic Reaction for Reactive Oxygen Species-Responsive Amphiphilic Diblock Copolymer”. We also thank you for your critical and professional comments for improving the quality of the manuscript. We have made specific efforts to perform additional experiments according to the comments/suggestions, and the manuscript has been revised adequately based on the supplemental results and a careful literature survey. The questions put forward by the reviewers have been answered point to point below.

Sincerely

Xiangqiang Pan

Round  2

Reviewer 1 Report

The authors  report 'Actually, the efficient of reaction decreased, and the conversion dropped down to 85% when the PEGMA 2000 was used. After purification process, the copolymer was also used for making the micelle. However, the micelle is not so stable for self-assembly. At last, we found PSSe-1-b-PEGMA950 was the best copolymer for self-assembly micelles in the current work. '

Where the experimental data with PEG 200 are reported?

Author Response

The response to Reviewer 2 at point 2 was inexact. Actually, the conversion of PSSe-1-b-PEGMA2000 is very low according to 1H NMR calculation (Figure 1). The coupling product (PS-SeSe-PS) was formed (Facile synthesis of well-defined redox responsive diselenide-labeled polymers via organoselenium-mediated CRP and aminolysis. Polym. Chem. 2015, 6, 1367-1372.). Furthermore, the copolymer cannot be purified thoroughly leading to a non-normal distribution SEC curve (Figure 2). As shown in the TEM image, large aggregates was produced because of oleophilic polymers (PS-SeSe-PS) in the copolymers (Figure 3).

Reviewer 2 Report

I think the author addressed all my previous comments. To improve the quality, please add the statistical analysis for the in vitro study. Also, there are typos in the paper. Please correct them.

Author Response

Thank you very much for giving us a chance to improve the quality of our manuscript, and we have added the statistical analysis for the in vitro study. Furthermore, the oxidation-responsive drug release of micelles was also added.

Round  3

Reviewer 1 Report

 I'm satisfied with the authors corrections